# Gelatin and Chitosan as Meat By-Products and Their Recent Applications

**DOI:** 10.3390/foods12010060

**Published:** 2022-12-22

**Authors:** M. Abd Elgadir, Abdalbasit Adam Mariod

**Affiliations:** 1Department of Food Science & Human Nutrition, College of Agriculture and Veterinary Medicine, Qassim University, Buraydah 51452, Saudi Arabia; 2Department of Biology, College of Science and Arts, Alkamil Branch, University of Jeddah, Alkamil 21931, Saudi Arabia; 3Indigenous Knowledge and Heritage Centre, Ghibaish College of Science and Technology, Ghibaish P.O. Box 100, Sudan

**Keywords:** gelatin, chitosan meat-by product, biological activities, antioxidant, edible coating

## Abstract

Meat by-products such as bones, skin, horns, hooves, feet, skull, etc., are produced from slaughtered mammals. Innovative solutions are very important to achieving sustainability and obtaining the added value of meat by-products with the least impact on the environment. Gelatin, which is obtained from products high in collagen, such as dried skin and bones, is used in food processing, and pharmaceuticals. Chitosan is derived from chitin and is well recognized as an edible polymer. It is a natural product that is non-toxic and environmentally friendly. Recently, chitosan has attracted researchers’ interests due to its biological activities, including antimicrobial, antitumor, and antioxidant properties. In this review, article, we highlighted the recent available information on the application of gelatin and chitosan as antioxidants, antimicrobials, food edible coating, enzyme immobilization, biologically active compound encapsulation, water treatment, and cancer diagnosis.

## 1. Introduction

Among the many meat by-products studied in the literature, gelatin and chitosan are the most abundant polysaccharides in nature [1]. Their use has attracted attention because they possess antimicrobial, non-toxic, and antifungal properties [2,3,4]. They are considered perfect animal by-product materials for the application and development of many products [5]. Gelatin is a protein that is obtained from the processing of animal bones and connective tissues. It is a colorless gel that cracks when dried due to the breakdown of collagen in tissues and bones. Gelatin is used in the food, pharmaceutical, and cosmetic industries [6].

Gelatin is a natural ingredient derived from animal by-products such as cattle bones, pork skins, and split cattle hides. It has healthy properties and has many applications, such as in confectionery, pharmaceutical products, meat, cosmetic and health care products, desserts, dairy products, and juices [7]. Chitosan is a natural polysaccharide that is created by deacetylating chitin (poly (-(1 4)-Nacetyl-D-glucosamine) [8]. Chitosan is a commercially available and cheap polysaccharide that is semi-crystalline and most commonly solvable in weak organic acids, such as lactic, acetic, formic, citric, tartaric, and malic acids [9,10]. Chitosan is synthesized by deacetylation of chitin (poly (β-(1 → 4)-Nacetyl-D-glucosamine), a natural polysaccharide [11]. It is a reasonably priced and easily accessible polysaccharide. In the solid phase, chitosan presents as semi-crystalline and soluble in dilute organic acid such as lactic, citric, acetic, formic, malic, and tartaric acids [12]. Gelatin and chitosan became more popular because of their applications in various industrial fields [13,14]. Chitosan and chitosan nanoparticles are considered promising bio-based polymeric materials and have received much attention in the medical and pharmaceutical fields in recent decades. Their applications are mainly focused on biomedical and tissue engineering fields [15,16,17]. They have extensive potential for application in the pharmaceutical [18] and medical fields as nanocarriers for drugs or active compounds that encapsulate substances, provide a controlled release, and deliver them to a specific place or site [19]. Several research studies have extensively investigated chitosan applications in various medicine and pharmaceutics fields. They are biocompatible and allow encapsulation of drugs and active medical ingredients [20]. In addition, chitosan and gelatin have other excellent properties, such as reducing damage to non-targeted cells or tissues and preventing the enzymatic degradation of drugs [21,22,23]. These properties make gelatin and chitosan great materials in the fields of cancer treatment, biological imaging and diagnostics, and drug delivery systems [24,25,26]. Furthermore, the slow biodegradation of gelatin and chitosan nanoparticles has been reported to ensure controlled and continuous drug release due to the highly positive surface charges that provide stable carriers that transport substances to the target position of the human body [27,28]. Recently, many studies have been conducted with the aim of using gelatin and chitosan as important animal by-products in various industrial trends [29,30,31,32,33,34,35,36,37,38,39,40,41]. The aim of this review is to summarize the recent advances in the applications of gelatin and chitosan as animal by-products in food, pharmaceutical, and medical fields. However, prospects for the application of gelatin and chitosan are also highlighted.

## 2. Properties of Gelatin and Chitosan 

Gelatin is mainly derived from untanned bovine hide wastes or by-products, fish skins, pigs, and poultry. It can be used in packaging and coatings, whereas chromium-tanned leather waste can be used in agriculture as fertilizer [42]. Gelatin is widely used in the food, pharmaceutical, cosmetic, and photographic sectors because it has special functional qualities. Gelatin is a food additive that is used in the bread, dairy, beverage, and confectionary industries to provide gelling, stability, texturization, and emulsification [43]. Chitosan, a natural linear bio-polyaminosaccharide (Figure 1), is produced from the major constituent of the protective cuticle, chitin, which is alkaline deacetylates in various crustaceans such as shrimp, crab, and lobster [44,45]. Chitosan is biodegradable, cheap, and non-toxic to mammals, which makes it applicable for use as an additive in the food industry [46]. Chitosan is a weak base and insoluble in water and organic solvents [47,48]. It forms a gel when it precipitates in an alkaline medium or with polyanions at low pH level [49]. Because of chitosan’s biodegradability and biocompatibility, it has been studied in a wide range of applications [50,51,52].

Collagen (Figure 2) is well known to be the most abundant protein in animals and is recognized as being the main component in leather. Due to its excellent bioactivity, high biocompatibility, and low antigenicity, collagen is a protein material that finds utility across many sectors, including those of medicine, leather, and cinema [53]. Although collagen is employed in the construction of scaffolds as well, it is always crucial to protect the bonding points throughout the collagen chain to prevent the loss of biocompatibility. Collagen has also been used to insert dental implants, speed up healing, and treat oral wounds by allowing cells with the capacity to regenerate to repopulate the damaged area [54]. Additionally, collagen casings are used to package meat and meat-related items. With the aid of nanotechnology and other strengthening processes, the case is formed of skin-derived collagen fibers [55].

Animal-derived collagen is used in the food industry, as well as gelatin in its denatured form which, traditionally, is derived as a by-product of meat rendering from connective tissue. In the food industry, gelatin is used to form gels, gums, emulsifiers and as a polyelectrolyte to aid flocculation.

### 2.1. Gelatin and Chitosan as Good Antioxidant

It is recognized that oxidation is one of the most significant problems in the quality of food products, and during the high-temperature processing of protein foods, heterocyclic amines are generated, which are known as carcinogenic substances [57]. Some factors, such as processing conditions, the presence of antioxidants, cooking methods, time, and temperature may influence the production of heterocyclic amines, and therefore, the reduction or inhibition of the formation of these carcinogens, has become an important issue [58]. The gelatin extracted from skipjack tuna (*Katsuwonus pelamis*) canning by-products was purified to give nineteen peptides that showed a high level of antioxidant activity. A high concentration of amino acids gives the gel exceptional clarity and strength. These results indicated that the antioxidant peptides generated from this gelatin might be used as possible additives in health-beneficial goods to prevent ultraviolet-A injury [59]. Chitosan added to food products as a food additive can act as an antioxidant agent. This prevents the formation of heterocyclic amines in foods [60]. Oz et al. [61] examined the impact of applying chitosan in concentrations of 0.25, 0.50, 0.75, and 1% w/w on the meatball’s quality and heterocyclic aromatic amine production. The meatballs were prepared at various temperatures (150, 200, and 250 °C). The results showed that increasing the temperature from 150 to 250 °C increased the content of heterocyclic amine in the meatballs. However, increasing chitosan concentration showed a significant decrease in the content of the heterocyclic amine. Similarly, Mirsadeghi et al. [62] showed that adding acid-soluble chitosan in the concentration of 1% to Huso fillets during cooking effectively reduced the production of heterocyclic amines and had an inhibitory effect of 68.09%. The antioxidant properties of an edible chitosan–galactose complex were investigated by combining chitosan and galactose (0, 0.5, 1, and 1.5 g). An in vitro test was also performed to evaluate the coating and determine the parameters for measuring antioxidant activity using the DPPH (2,2-diphenyl-1-picryl-hydrazyl-hydrate) method [63]. The IC50 values decreased slightly with increasing amount. The strongest antioxidant in the treatments, a mixture of chitosan and 1 g galactose, had the lowest IC50 value of 43.20 ppm. Similarly, Zhang et al. [64] published the findings of studies using chitosan with high molecular mass for radical scavenging activity, demonstrating that high-molecular-weight chitosan had strong hydroxyl radical scavenging, while low-molecular-weight chitosan had higher superoxide anion radical and DPPH scavenging. Adiletta et al. [65] evaluated the activities of enzymes such as polyphenol oxidase and peroxidase, catalase, and ascorbate peroxidase to determine the impact of the chitosan-based coatings on the freshness of figs. The results revealed that the addition of chitosan coating significantly increased the flavonoids, anthocyanins, and total polyphenol contents and antioxidant activity of the stored figs, reduced oxidative stress, and prevented browning reactions compared to the untreated group. The antioxidant capacity and shelf life of strawberries were studied by Martínez-González et al. [66] to assess the impact of the propolis/nano-chitosan coating on the film-coated strawberries. At the end of the preservation period, the strawberries in the nano-chitosan/propolis-coated group had greater overall phenolic and flavonoid levels and antioxidant capacity than the strawberries in the untreated group. Because chitosan lacks hydrogen atoms that can be readily added to make it a potent antioxidant, chitosan-based basic films have relatively little ability to scavenge free radicals [67]. Several naturally occurring active substances, such as volatile oils, green or black tea extracts [68], apple extract [69], banana peel extract [67], purple and black eggplant extracts [70], and black and purple rice extracts [71] can increase the antioxidant capabilities of chitosan. This phenomenon has been attributed to the fact that the polyphenols contained in the extract can scavenge free radicals by releasing phenolic hydrogen atoms [72]. 

### 2.2. Gelatin and Chitosan as Antimicrobial

Kavoosi et al. [73] discovered that gelatin films infused with thymol had extremely potent antibacterial properties, making them suitable for use as antibacterial nanowound dressings against pathogens that cause wound burns. This makes them suitable for use as antibacterial nano wound dressings against pathogens caused wound burns [74]. They absorb exudates, sustain a moist environment on the wound surface and imitate the extracellular matrix structure and have an antibacterial effect [75]. Because gelatin films with bergamot and lemongrass essential oils have good antibacterial properties and display better heat stability with higher breakdown temperatures, they can be employed as active packaging materials [76]. Chitosan is a cheap and non-toxic compound; it is also used as an antifungal in agriculture, as a food additive in the food industry, and as a wetting agent in cosmetics, in addition to its use in the synthesis of some medicines in biomedicine [77]. Chitosan nanoparticles and liposomes containing ethanolic cinnamon extract were prepared by Elwakil et al. [78]. They studied their physical and chemical properties before determining how well they healed wounds. They created a gel using chitosan and liposomes that contained ethanolic cinnamon extract and tried it on diabetic mice. They discovered that treating bacterial infections and blocking enzymes required the liposome/cinnamon gel to be more successful. Chitosan is more efficient against Gram-positive bacteria than Gram-negative bacteria, as demonstrated by earlier studies, and can inhibit the growth of a variety of bacteria and fungus [79]. The use of chitosan and essential oil formulation in chitosan-based edible packaging films increased the effectiveness of antimicrobials against gram-negative bacteria, including *Escherichia coli* [80], *Pseudomonas aeruginosa* [81], *Pseudomonas fluorescens* [82], *Klebsiella pneumoniae* [83], *Shewanella putrefaciens*, *Shewanella baltica*, *Serratia spp*. and Gram-positive bacteria such as *Staphylococcus saprophyticus* [84] and *Staphylococcus aureus* [85]. However, yeast, fungus, and mold are also inhibited [86]. Chitosan sheets were tested against *Penicillium italicum* in combination with bergamot essential oil and showed a great inhibitory effect, but the inhibitory potency of the composite sheets decreased during the storage period [87]. The volatile oils of cinnamon inhibited the growth of Aspergillus oryzae, Botrytis cinerea, Aspergillus niger, Penicillium digitatum, and Rhizopus stolonifera fungi on chitosan films [88,89]. Li et al. [90] observed that the use of essential oil of turmeric in chitosan resulted in significant anti-aflatoxigenic activity thanks to the observed antifungal properties against *Aspergillus flavus*. The application of *Eucalyptus globulus* essential oil–chitosan matrix successfully inhibited yeasts such as *Candida parapsilosis* and *Candida albicans* [81]. Chitosan was also studied when incorporated into extracts of polyphenols to enhance its antimicrobial properties and such polyphenol include pomegranate peel extract [91], green tea extracts [92], spirulina extract [93], propolis extract [94], black plum peel extract [64], and purple corn extract [95]. It was claimed that the powerful antimicrobial activity of essential oils when incorporated in chitosan was because they contain terpenes, which affect bacterial membrane permeability in addition to various functions and cause the death of bacterial cells by raising the amount of lipid peroxides such as alkoxyl, alkoperxyl, and hydroxyl radicals [96]. The blended films of gelatin and chitosan showed good antioxidant properties in the Trolox equivalent antioxidant capacity assay test and incredible growth suppression against Staphylococcus aureus and Escherichia coli, indicating that such blends’ ethanolic extract sensitivities could provide a substitute as effective packing for applications in the food industry [97]. According to Kurczewska, [98], gelatin and chitosan, as well as their derivatives, are biodegradable polysaccharides that are biocompatible, non-toxic, and have antimicrobial and antifungal properties. Recently, they have been used in intelligent food packaging as active ingredients to enable shelf life to be extended and guarantee quality and food safety [64]. They act as active scavenger systems, including oxygen scavengers, moisture scavengers, and ethylene scavengers [99], leading to an extension of shelf life and preservation of food quality [100]. Chitosan–polyphenol extract was investigated against Gram-negative bacteria such as *Salmonella typhimurium, E. coli, Proteus mirabilis, Salmonella enterica, P. aeruginosa,* and *Proteus vulgaris* and showed significant antibacterial activities [93,101]. Chitosan–polyphenol extracts also demonstrated effective antibacterial activity against Gram-positive bacteria such as *Streptococcus mutans*, *S. aureus*, *Bacillus cereus*, *Bacillus subtilis*, *Listeria monocytogenes*, *Bacillus thuringiensis*, *Lactobacillus sakei*, *Lactobacillus plantarum*, and *Listeria innocua* [64,92,93,102].

### 2.3. Gelatin and Chitosan as Food Edible Coating Source

Recently, gelatin and chitosan have been used in food packaging because the use of petroleum-based materials has detrimental effects on the environment because they are not sustainably sourced, reusable, recyclable, or renewable [103,104]. Research on food packaging must address the environmental problems caused by the careless use and handling of non-biodegradable components and provide new, environmentally friendly options. Biodegradable natural polymers that have been investigated for potential uses in the food packaging sector, among them chitosan and gelatin [105], have attracted a great deal of interest in recent decades. Active films made of 15% gelatin, 30% glycerol, and 1% green tea extract were prepared by Hamann et al. [106]. These films were added to the fresh sausages’ coating. Their findings demonstrated that during cold storage, TBARS levels in sausages coated with active gelatin film were reduced. Finally, they concluded that gelatin films infused with green tea extract are a promising substitute for extending sausages’ shelf lives [106]. Dehghani et al. [107] produced coating dispersions with fish gelatin, conjugates, or bitter almond gum (1:2, 2:1, 1:1). They looked at how the coating suspensions affected the physicochemical and qualitative indicators of tomatoes stored at 20 °C for 28 days. These authors found that the conjugation of fish gelatin with a higher bitter almond gum ratio could be promising for producing coating dispersion and maintaining fruit quality during storability. According to a study by Jusoh et al. [108], virgin coconut oil can be used with gelatin film to create active film packaging or edible film packaging for some culinary applications, such as packing material for protein-rich foods like meat. Singh et al. [109] created chitosan-based films with oxygen-scavenging capabilities by incorporating sodium carbonate and gallic acid into the polymer chain. The incorporation of TiO_2_ nanoparticles in chitosan sheets imparts ethylene-scavenging properties [110]. Chitosan-based smart films were developed by Nandeesh and Kalpana [111], including two main groups of chitosan smart packages: (1) sophisticated biosensors; and (2) films with a visual color change due to colorimetric reactions. These packages include time–temperature indicators, pH indicators, and freshness indicators. Nevertheless, Wang et al. [112] employed a chitosan–gold nanoparticle combination to show the frozen state and temperature history of food through the color difference that appears when gold nanoparticles clump together because of their localized surface plasmon resonance. Additionally, because of the physicochemical changes in the food, chitosan-based materials designed to monitor pH variations in food can also identify bacterial load and oxidative food deterioration. Singh et al. [109] added sodium carbonate and gallic acid to the chitosan film to develop oxygen-scavenging material. The results showed a decrease in mechanical parameters of the chitosan films as the concentration of the added sodium carbonate and gallic acid increased. This may be due to the large amount of sodium carbonate disrupting the inner matrix of the chitosan film [102]. Another use of chitosan in food packaging is as humidity sensors, which are based on chitosan-zinc oxide, and single-walled carbon nanotubes. The chitosan swelling impact that surrounds the nanotubes in this usage is thought to be the sensing mechanism, altering the hopping conduction channel between nanotubes [113]. Zhang et al. [67] developed moisture sensors based on a quartz crystal microbalance coated with chitosan multi-walled carbon nanotubes. The optimized sensor can be used to detect food moisture with the features of negligible humidity hysteresis, high response sensitivity, fast response and recovery times, repeatability, remarkable reversibility, and long-term stability and selectivity. However, the addition of quercetin to chitosan films enables the intelligent detection of aluminum (Al 3+) in food based on colorimetric reactions [114], because quercetin can form bonds with Al 3+, resulting in a colored complex. A graphene oxide/chitosan nanocomposite-coated quartz crystal microbalance sensor for the detection of amine vapors was investigated [64]. The sensor displayed high aliphatic amine sensitivity at ambient temperature, containing methylamine, dimethylamine, and trimethylamine. Although there are instances of controlling CO_2_ production by developing a pH-CO_2_-generated link, these substances are primarily used as markers of food pH and freshness. The most significant category of flavonoids and a significant component of phenolic compounds is anthocyanins. These dyes exhibit color alterations in response to pH variations. However, further research is required before chitosan-based biosensors can be used in intelligent food packaging. Depending on variations in impedance, sophisticated humidity and temperature sensors were created using chitosan and CuMn2O4 spinel nanopowder. The reduction in the sensor’s impedance with rising temperature is due to charge carrier production, which is influenced by temperature [115]. Research analyzing active and smart materials having both anthocyanins’ capabilities is frequently found, as they also have powerful antioxidant effects. For packaging chicken breasts at 4 °C, a curcumin-loaded chitosan and polyethylene oxide nanofiber film was created as a freshness marker. The nanofiber film’s hue altered from light yellow to reddish, enabling even the inexperienced consumer’s naked eye to identify color variations [116]. El-Gioushy et al. [117] studied nano-chitosan as an active edible coating film in concentrations of 1, 2, and 3 cm3/L for enhancing the shelf life and quality properties of date palm fruits (Barhi cultivar) during cold storage at ±2 ℃ for 70 days and discovered that at the end of the storage period, spraying the Barhi date fruit with 3 cm3/L of nano-chitosan achieved the best results. The usage of chitosan in enriched chitosan packing films has been found to have worse mechanical resistance characteristics than pure chitosan treated samples, including lower values of percent elongation at break (%E) and tensile strength (TS) [118]. This effect was observed with the addition of essential oils such as *Artemisia campestris* [118], *Perilla frustescens* [119], basil [84], ginger (*Pimpinella anisum* L.) [120], and *Artemisia campestris* [118]. The addition of polyphenol-rich extracts to chitosan-based films, such as green tea extracts [121], apple extracts [69], banana peel extract [67], Chinese chives [122], root extract [123], mango kernel extract (honeysuckle flower extract [124], *Pistacia terebinthus* leaf extract [125], syringic acid, and purple pulp sweet potato extract [126], protocatechuic acid [127], resulted in an overall trend of decreasing TS and % E values.

### 2.4. Gelatin and Chitosan in Microencapsulation Technology

Microencapsulation technology is used to prevent product base materials from deteriorating by enhancing the active components’ bioavailability, which increases their solubility and enables the preparation of solid formulations of oils. The efficiency of the capsule is determined by the properties of the wall and the base materials. Excellent results can be obtained by using the mixture of the wall material to prepare the microcapsules. Rosmarinic acid and carvacrol, the two main active components in Turkish oregano extract, have been found to release more readily in vitro when gelatin, gum arabic, Tween 20, and cyclodextrin were used as coating materials [128]. Chitosan’s qualities make it an ideal coating material for encasing a variety of bioactive substances. This makes it useful in the biomedical, food, agricultural, pharmaceutical, environmental, and industrial fields [129]. This polymer is used to encapsulate food ingredients, essential oils, vitamins, lipids, drugs, vaccines, microbial metabolites, and hemoglobin [130]. Chitosan and its encapsulated compounds are widely used in agriculture in some ecological alternative products such as organic fertilizers, biopesticides, soil conditioners, seed treatment, and growth promoters’ agents [131]. Chitosan has been used as a co-encapsulation material for resveratrol and curcumin [132]. Chitosan is also used in the development of nanocomposite active compounds in films to inhibit the growth of fungi such as *Aspergillus flavus*, *Aspergillus parasiticus*, *Aspergillus niger*, and *Penicillium chrysogenum*, resulting in the control and inhibition of these pathogens [133].

### 2.5. Gelatin and Chitosan in Water Treatment

Heavy metals such as copper, nickel, lead, zinc, cadmium, mercury, arsenic, chromium, bismuth, cobalt, and iron are harmful to the environment and human health even when present in trace amounts [134]. Eliminating these heavy metals from wastewater is of paramount importance, as they not only pollute water bodies, but are also toxic to the ecosystem [135]. Gelatin has been combined with yeast to create the GelYst biosorbent, which is used to improve the extraction and biosorption of Cr (VI) from water. This biosorbent’s applications in water treatment have been successful [136]. Chitosan is used as an inexpensive dye remover and heavy metal biopolymer [137]. Compared to other commercial adsorbents, chitosan has received much attention in water treatment applications due to its specific properties, such as high adsorption capacity, cationicity, macromolecular structure, low price, and abundance [76]. Various metals and other pollutants have been reported to be effectively removed by chitosan or various modifications of this biopolymer [138].

### 2.6. Gelatin and Chitosan in Tissue Engineering

Gelatin methacryloyl (GelMA) hydrogels with cell-responsive arginylglycyl aspartic acid and matrix metalloproteinases peptide sequences have been frequently used in tissue engineering because of their adaptable mechanical, superior processing performance, and outstanding biocompatibility properties. GelMA-based hydrogel microstructures can be precisely controlled using modern production techniques such as 3D printing and electrospinning. GelMA hydrogels with different microstructures have been designed and studied to mimic the natural extracellular matrix and to control the proliferation, migration and differentiation of different cell types [139]. Chitosan can act as an ideal agent for wound dressing due to its positive charge and mild gelation properties, film-forming ability, and strong tissue-adherent properties with improved blood coagulation [140]. It supports wound healing by increasing the functions of inflammatory cells such as polymorphic nuclear leukocytes, macrophages, and fibroblasts [141]. Chitosan also has potential use in skin repair and regeneration after injury or burns, as it can be cross-linked with silica (SiO_2_) particles. It was found to be non-cytotoxic to L-929 cell culture when used in extraction forms in engineered membranes. Furthermore, the macroporous membrane showed excellent cell adhesion and proliferation after 24 h and 48 h of cultivation [142]. Chitosan-based materials have also been shown to have the potential to maintain and stimulate cell phenotypes [143].

### 2.7. Gelatin and Chitosan in Drug Delivery

The potential use of chitosan and gelatin as drug delivery carriers has been reported in several studies [144,145,146]. Gelatin was applied to increase the efficiency of drug delivery into cancer cells by coating drug-encapsulating liposomes with gelatin [22]. The investigated liposomes were coated with gelatin using electrostatic interaction and covalent bonding methods. The coated drug was compared with polyethylene glycol liposomes in terms of encapsulation efficiency, size, stability, zeta potential, cell uptake, and dissolution profile. The results showed high drug-encapsulation efficiency and sustained release depending on the degree of gelatin coating. The cell uptake studies showed that the gelatin-coated liposomes were superior to polyethylene glycol liposomes in terms of cancer cell targeting ability. Alginate, chitosan, pullulan, and their combination nanoemulsions were developed, optimized, and characterized by Fard and his research team [147] as promising drug delivery platforms for melanoma. A unique nanoemulsion delivery method was developed, and its effectiveness was evaluated using confocal microscopy, in vitro drug release, cell survival, and cellular death. The results obtained show the significance of the polymeric mixture of the drug carrier and the effect of the drug release pattern on the effectiveness of the therapy.

### 2.8. Summary of Gelatin and Chitosan Potential Applications 

Gelatin and chitosan could be used in many industrial fields, including in foods, medicines, and pharmaceuticals (Table 1 and Table 2). They have been used to replace disposable plastic packaging materials that pollute the environment [148]. They can be used to create biodegradable packaging materials for use in favor of plastic packaging materials. Furthermore, inorganic nanoparticles of some materials, like silica, metal, and carbon nanomaterials, have been studied and have shown successful applications in the field of nanocomposites [149,150]. They can be incorporated into biodegradable packaging materials to improve their quality. In addition, gelatin and chitosan possess powerful properties such as antimicrobial and antioxidant activities that could help extend the shelf life of the packaged materials [151]. Chitosan and gelatin contain hydroxyl and amino functional groups, making them interesting materials for removing a wide range of pollutants, such as pesticides, dyes, and heavy metals [152]. 

## 3. Conclusions

In this review, we highlighted recently available information about chitosan and gelatin and their applications in food, medicine, pharmaceuticals, and wastewater treatment. Gelatin and chitosan are attracting great interest due to their numerous applications in food, medicine and pharmaceutical products. They are interesting biomaterials for product development of widespread application due to their excellent inherent properties including biodegradability and food ability nature. With the development of technology and the economy, gelatin and chitosan are often produced according to legislation or standards that consider safety and health. Gelatin derived from meat by-products is an ideal material for food packaging due to its many advantages such as low price, polymerization, biodegradability, and good antibacterial and antioxidant properties. However, the gelatin film has poor water and mechanical resistance, which is an obstacle to its development and application in food packaging. The application of gelatin and chitosan as alternative biomaterials for edible biodegradable polymers for packaging is considered to be one of the best ways to reduce plastic pollution in the environment.

## Figures and Tables

**Figure 1 foods-12-00060-f001:**
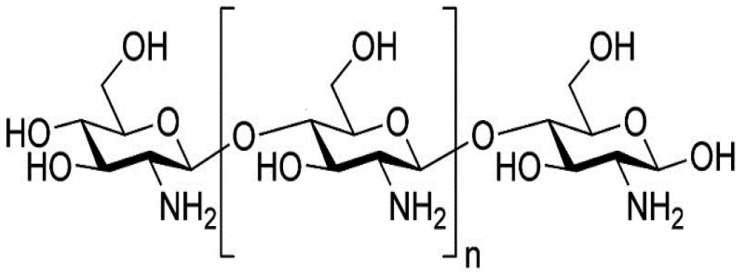
Structure of chitosan.

**Figure 2 foods-12-00060-f002:**
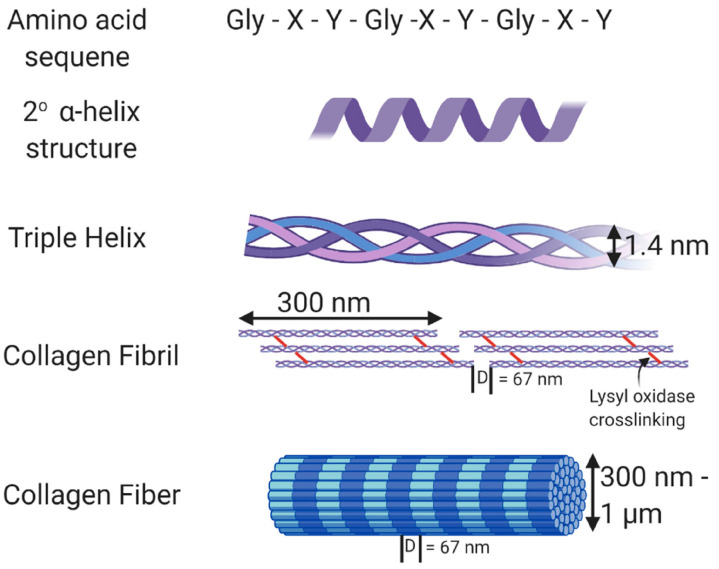
Structure of collagen [56].

**Table 1 foods-12-00060-t001:** Selected applications of chitosan in the food industry.

No	Type of Gelatin & Chitosan Application	Purpose of the Application	References
1	Edible coating	Limit oxygen uptake, reduce moisture transmission, retard ethylene production, reduce respiration, exhibit selective gas permeability, exhibit resistance to fat diffusion, and trap volatile flavor compounds.	[107]
2	Emulsifier agents	Gelatin is an important natural amphiphilic macromolecule and can act as an emulsifier in oil-in-water emulsions due to its surface-active properties. Chitosan nanoparticles as a special emulsifier for the production of stimulus-dependent Pickering emulsions—oil-in-water and stable emulsions.	[153]
3	Shelf-life extension	Chitosan/pectin multilayer packs increase the shelf life of tomatoes. Chitosan coating papaya and mango slices also helped extend their shelf life by delaying water loss and a drop in sensory quality. The tomato conjugate with bitter almond gum and fish gelatin (2:1) received higher impression and sensory scores.	[154]
4	Antioxidant	Inhibition of DPPH, ABTS, and their protective effects on human erythrocytes. Gelatin and antioxidant peptides derived from it could serve as potential ingredients to be used in health-promoting products to prevent ultraviolet-A injury.	[155]
5	Food biosensors	Electrochemical chitosan-based biosensor: application for the determination of acrylamide in food product samples. In the realms of food testing, medical diagnosis, and environmental control, gelatin with biosensors is frequently employed for the detection of numerous analytes such as glucose, urea, amino acids, hydrogen peroxide, and pesticides.	[156,157]
6	Antimicrobial	Gelatin films integrated with thymol are very powerful antimicrobial. Thus, they can be used as antibacterial nanowound dressings against pathogens that cause wound burns. Chitosan micro and nanoparticles have been shown to be effective against pathogens such as *E*. *coli O157: H7*, *S. enterica*, *E. coli*, *Klebsiella pneumoniae*, *Vibrio cholerae*, S. *choleraesuis*, *Streptococcus uberis*, and *S. aureus.*	[99]
7	Anticholesterol agents	Chitosan is used to reduce low-density lipoprotein (LDL) cholesterol levels and its fat-binding capacity.	[158]
8	Food additive	In the food industry, gelatin is used as a gelling, thickening, or stabilizing agent. Chitosan-grafted hydrogels are useful for those with lactose intolerance due to the controlled release of galactosidase.	[43]
9	Diet Supplementary	Dietary supplementation with pyridoxine-loaded vanillic acid-grafted chitosan microspheres and thiamine enhances metabolic, immune, and growth performance responses in experimental rats. Gelatin may be used as a supplement to reduce the risk of osteoporosis, or a thinning of the bones.	[159,160]
10	Purification of water	A gelatinous yeast biosorbent is used to improve the extraction and biosorption of Cr(VI) from water. Removal of metal ions, phenols, pesticides, dyes, PCBs, DTT, amino acids, proteins, oil, and greases.	[161]

**Table 2 foods-12-00060-t002:** Selected other applications of chitosan in different fields.

No	Application Area	Benefit of Application	References
1	Regeneration technology	Chitosan can be used in neural regenerative technology, bone regeneration, cardiac regeneration therapy, corneal regeneration technology, and skin regeneration technology.	
2	Immune therapy	Chitosan can activate CD 4+ cells, the complement system, and humoral immunity.	[162]
3	Environmental protection	Chitosan is used in the removal of various inorganic and organic pollutants from the environment, as well as heavy metals and harmful pesticides	[163]
4	Drug absorption enhancer	The Caco-2 and intestinal cells can take in more drugs with the help of hydrogel-based superporous systems and other chitosan complexes.	[164]
5	Cells immobilization	Cells such as E. coli can be immobilized by the application of chitosan beads.	[165]
6	Gene therapy	Delivering various genes that are used in gene therapy, siRNA and cancer therapy technology	[163]
7	Paper manufacture	Production of water-resistant papers, biodegrading packages, and filter papers.	[166]
8	Energy production	Chitosan can provide ionic conductivity in an acetic acid solution and can be used in the production of solid-state batteries.	[167]
9	Cosmetics	Due to its UV absorbing ability, biocompatibility, antifungicidal activities, and chitosan can be used in various cosmetic industries.	[14]
10	Wood industries	Chitosan is used as a wood quality enhancer, wood adhesive, and antifungal agent.	[168]

## Data Availability

Not applicable.

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
