# Peer review of "Gelatin and Chitosan as Meat By-Products and Their Recent Applications"

_foods, 2022, doi:10.3390/foods12010060_

Round 1

Reviewer 1 Report

A native English speaker should correct the many language and grammar difficulties. There was a lack of organization of information on the two by-products in each section and a discussion of similarities and differences in structures of each would give needed explanations of their functions. There did not seem to be added narrative, explanation, or insight into the information not found in the many review articles that were cited.

Line(s)      Comment

43-45        Sentence should be reworded to correct grammar error.

72              Section should be “Properties of gelatin and chitosan” since the subsections describe the properties and not necessarily the applications of these materials.

92              Additional information about the structures of collagen should be included for consistency because the structure of chitosan is in Figure 1.

95              Section should be “Gelatin and chitosan antioxidant properties” because “good” is a subjective term and implies a comparison will be made.

107-110   This sentence about gelatin films in packaging should be in section 2.3.

111-112   The mention of microbial growth suppression should be moved to section 2.2.

146-150   This sentence must be rewritten for improved clarity of its meaning.

153           “antimicrobials”

155-156   Clarification is needed that pathogens cause infection in wound burns.

197-200   The sentence should be reworded so that the chitosan-polyphenol extract and not the Gram-negative bacteria would be indicated to provide significant antibacterial activities.

206           This section includes both coatings and inclusion into packaging materials so the section title should be renamed to indicate this or, preferably, another section on incorporation of gelatin and chitosan into packaging materials be inserted that contains this information from the current section 2.3.

212-215   Poorly worded sentence.

225-232   The information in these sentences should be incorporated into l. 207-215.

232-236   Move to l. 43.

236-238   Delete as unnecessary information.

307-308   Incomplete sentence.

281-392   This section does not suggest future applications that weren’t previously discussed. The section title could be “Summary of gelatin and chitosan potential applications.”

395-400   The table are not mentioned in the text. Because they summarize information in the text, their value is questionable without additional explanation.

401-406   Include this information in section 2.8 and relabel as 3.0.

457-459   Spacing inconsistency.

505-506   Title format is inconsistent with titles of other references.

543-545   Different font size than other references.

560           Journal name is not italicized in other references.

566           Citation is incomplete.

615-617   Repetition of the previous reference.

628-629   Repetition of the previous reference.

680-682   Different font size than other references and repetition of the previous reference.

669           Journal name is not italicized in other references.

748-749   Repetition of the previous reference.

751           Journal title not italicized in other references.

Author Response

Dear reviewer

Thank you for your valuable comments which I respond to point by point  

Reviewer 2 Report

This was studied to Gelatin and chitosan as meat by-products and their recent applications, however, the language is not good. Especially, the structure of this paper is disordered, need to rewritten. There are some question need answer:

1. Abstract is not summarize the result of this paper, need rewrite.

2. The format of the article needs a complete overhaul.

3. Introduction is so simply, need rewrite.

4. The aim of the article is not clear, please explain the innovativeness.

5. The conclusion of “2. Application of gelatin and chitosan”?

6. Figure 1. Structure of chitosan is not show, why?

7. L 140  (2020to?

8. L154-155 The sentence need rewrite.

9. L159 Ahmed, et al. 2012 or Ahmed, et al., 2012?..................more mistakes like this.

10. The conclusion of “2.3. Gelatin and chitosan as food edible coating source”?

11. L265 (Al3+) or (Al3+)? ..................more mistakes like this.
12. 2.5. Gelatin and chitosan in water treatment: please rewrite.

13. L354-361 The sentence need rewrite.

14. 2. 8. Future perspective applications of gelatin and chitosan need rewrite.

15. The format of Tables are not true.

Author Response

(The authors gave the same response as above.)

Round 2

Reviewer 1 Report

Consistency is needed in "gelatin" spelling. 

Author Response

1st Reviewer

No

1

Minor spell check required

Spell check was done all through the manuscript

2

Consistency is needed in "gelatin" spelling

Consistency in “gelatin" spelling is considered all through the manuscript

Reviewer 2 Report

L 60-67 The sentences need improve.

2.6. Gelatin and chitosan in tissue engineering

The formation of sentences need improve.

3.0. Conclusions need improve. Especially L464-466.

All formats need to be modified as required in this paper.

Author Response

2nd Reviewer

L 60-67 The sentences need improve

The sentences were improved

2.6. Gelatin and chitosan in tissue engineering - the formation of sentences need improve

The formation of the sentences was improved

3.0. Conclusions need improve

Conclusions section was improved
